

# Different types of plantar vibration affect gait characteristics differently while walking on different inclines

Haoyu Xie[1], Haolan Liang[1] and Jung H. Chien[2]

[1] Department of Health & Rehabilitation Science, University of Nebraska Medical Center, Omaha, NE, United States
[2] Independent researcher, Omaha, NE, United States

## ABSTRACT

**Background:** Plantar vibration has been widely used to strengthen the sensation of the somatosensory system, further enhancing balance during walking on a level surface in patients with stroke. However, previous studies with plantar vibration only involved the level surface, which neglected the importance of inclined/declined walking in daily life. Thus, combining the plantar vibration and inclined/declined walking might answer a critical research question: whether different types of plantar vibration had different effects on gait characteristics during walking on different inclines.

**Methods:** Eighteen healthy young adults were recruited. Fifteen walking conditions were assigned randomly to these healthy adults (no, sub-, and supra-threshold plantar vibration × five different inclines: +15%, +8%, 0%, −8%, −15% grade). A motion capture system with eight cameras captured 12 retro-reflective markers and measured the stride time, stride length, step width, and respective variabilities.

**Results:** A significant interaction between vibration and inclination was observed in the stride time ($p < 0.0001$) and step width ($p = 0.015$). *Post hoc* comparisons found that supra-threshold vibration significantly decreased the stride time (−8%: $p < 0.001$; −15%: $p < 0.001$) while the sub-threshold vibration significantly increased the step width (−8%: $p = 0.036$) in comparison with no plantar vibration.

**Conclusions:** When walking downhill, any perceivable (supra-threshold) vibration on the plantar area decreased the stride time. Also, the increase in step width was observed by non-perceivable (sub-threshold) plantar vibration while walking uphill. These observations were crucial as follows: (1) applying sub-threshold plantar vibrations during uphill walking could increase the base of support, and (2) for those who may need challenges in locomotor training, applying supra-threshold vibration during downhill walking could reach this specific training goal.

Corresponding author
Jung H. Chien,
drjc.science@gmail.com

## INTRODUCTION

"If you want to find the secrets of the universe, think in terms of energy, frequency, and vibration" by Nikola Tesla (*AZ Quotes, 2020*). This vibration concept has been applied to enhance the system's ability to detect weak signals in humans and has been published in

Nature (*Collins, Chow & Imhoff, 1995*). Moreover, the concept of vibration has been further applied in clinical situations. A published review summarizes that vibration therapy with different frequencies and amplitudes demonstrates immediate and long-term effects on improving muscle strength, balance, and bone mass in healthy participants and patients with cerebral palsy (*Rauch, 2009*). Their results indicate the wide application and clinical benefits of vibration. Among different types of vibration, plantar vibration, applied to the sole, has been verified to improve haptic perception and neuromuscular activation not only in patients but also in healthy participants (*Layne, Forth & Abercromby, 2005*; *Layne & Forth, 2008*). The latest studies suggest that plantar vibration increases vertical jump height and decreases the minimal toe clearance variability in healthy young adults, indicating the beneficial role of plantar vibration in enhancing locomotor performance (*Moon et al., 2022*; *Pathak et al., 2022*). Specifically, based on different amplitude and frequency combinations, different rehabilitation protocols with vibration can be developed depending on the needs of clinical practice (*Murillo et al., 2014*).

Two common types of vibrations, sub-, and supra-threshold vibration, have been widely used to enhance postural control in standing and walking (*Acuña, Zunker & Thelen, 2019*). On the one hand, the effect of imperceptible sub-threshold vibration (90% of the amplitude of vibration that participants could perceive) can be explained by the concept of stochastic resonance. This concept refers to the phenomenon whereby the presence of vibration enhances the perception of weak sensory stimuli by reducing the threshold of somatosensory receptors (*Likens et al., 2020*). In other words, when applying the sub-threshold vibration on the sole of the foot, environmental information can be perceived from more somatosensory receptors. Thus, the interaction between humans and the environment can be more precisely estimated and controlled by the central nervous system to further enhance the static and dynamic balance in both healthy adults and patients with deterioration of somatosensory system (*Likens et al., 2020*; *Song et al., 2022*). Therefore, it has been generally concluded that applying the sub-threshold vibration to the human body enhances static and dynamic balance. On the other hand, perceptible supra-threshold vibration has been thought to play a role in perturbing the somatosensory system (*Song et al., 2022*). Because of the perturbations on the somatosensory system, the central nervous system might re-calibrate the gains from other sensory systems, such as the proprioceptive sensory system, to maintain balance under different locomotor tasks, such as obstacle negotiation (*Song et al., 2022*) and walking on a flat treadmill (*Pathak et al., 2022*). The direct evidence related to the abovementioned re-calibration is the alternations in gait characteristics and their variabilities (*Horak, 2006*; *Song et al., 2022*, *Pathak et al., 2022*).

Gait characteristics and their respective variabilities have been widely agreed on as indicators of dynamic balance control. For instance, while applying sub-threshold vibration on the sole, the stride time variability significantly decreases in recurrent fallers and non-fallers, and no significant difference in stride length and step width variabilities is observed (*Galica et al., 2009*; *Stephen et al., 2012*), inferring two-folded meanings: (1) applying sub-threshold vibration on the plantar area may have a more substantial impact on temporal domain than the spatial domain of gait characteristics and
gait variabilities, and (2) the decrement of variability indicates that recurrent fallers require less step-to-step adjustments and further reduce the potential risk of falls (*Maki, 1997*; *Lipsitz et al., 2015*). Similarly, applying a supra-threshold on the plantar area decreased the minimum toe clearance variability but not the mean values of the minimum toe clearance (*Pathak et al., 2022*). While performing a challenged locomotor task, stepping over an obstacle, the results show that both sub- and supra-threshold vibrations can significantly increase the toe clearance of leading and trailing legs during obstacle negotiation (*Song et al., 2022*). Importantly, their results also suggest that vibration has no effect on step time and step length (*Song et al., 2022*). The abovementioned observations may have three-folded meanings: (1) applying supra-threshold plantar vibrations may affect the gait patterns in both horizontal and vertical planes, (2) applying supra-threshold vibrations may interfere with the gait characteristics during other locomotor tasks other than walking on level surfaces, and (3) the variability measure also could be used to identify the alternations of gait patterns due to the deteriorations of sensory systems (*Maki, 1997*).

When considering the activities of daily living, walking up/downhill is inevitable. With the change in locomotor tasks and environmental demands, the body adjusts the gait strategy to keep balance and avoid falling while completing the inclination (*Jeon et al., 2020*). Similarly, from the complex behavior's hypothesis point of review, walking on different inclines extends the critical value in the first period-doubling routes to chaos, which are found in a passive walk of the compass-type bipedal robot model (*Added, Gritli & Belghith, 2021*). This alternation of the route to chaos has been thought of as a combination of changes in the location of the center of mass and the moment of inertia during walking on uneven terrains or the different inclines of surfaces (*Zhao et al., 2011*). In short, walking on different inclines induces different complexities of locomotion depending on the locomotor tasks and environmental demands.

Previous studies have shown that young and older adults may employ a cautious gait pattern with shorter step time (higher cadence), shorter step length, and broader step width during downhill walking to mitigate the risk of falling than walking on level ground (*Sample et al., 2020*; *Sheehan & Gottschall, 2015*). In contrast, increased step length and decreased cadence are observed when walking uphill (*McIntosh et al., 2006*). For the gait variabilities during walking on different inclines, the U shape is observed in the step length variability, the step time variability, and the step width variability, indicating that both uphill and downhill walking increase the gait variability in comparison with walking on the flat self-paced treadmill (*Castano & Huang, 2021*). Specifically, the step length variability is significantly greater during both uphill and downhill walking compared to level walking. Additionally, the step time variability is greater only during downhill walking compared to the other two walking conditions. For the step width variability, when walking either downhill or uphill, the results show a tendency to increase in the step width variability compared to walking on a level surface. Also, the measure of gait variability derived from non-linear dynamics, such as entropy, indicates that higher movement irregularity is observed when walking uphill than walking downhill in both young and older adults (*Vieira et al., 2017*). This observation suggests that uphill walking requires a higher level of degrees of freedom than downhill walking because uphill walking needs higher energy to

push the body forward. In contrast, downhill walking may require more caution to prevent tripping over the treadmill than uphill walking; thus, limiting the degree of freedom of movement might be inevitable to ensure walking safety (*Vieira et al., 2017*). Therefore, could interaction between the different types of vibration and levels of inclines on gait variability be found? This study attempted to answer this knowledge gap.

The findings from the abovementioned studies led to an important research question: whether different types of plantar vibration affected the gait characteristics and respective variabilities differently when performing other daily locomotor behaviors, such as walking on different inclines, including uphill and downhill. Therefore, this study aimed to answer the abovementioned question. To our best knowledge, this is the first study to investigate the effect of different types of plantar vibration on different gait characteristics during walking on different inclines. Understanding the alternations in these gait characteristics under the abovementioned sensory-conflicted combined inclined conditions could identify the priorities of these gait characteristics for future use in clinical applications. Importantly, this study may provide new ideas for the clinical application of vibrations and inclined treadmill training as rehabilitation protocols. We hypothesized that the interaction between different types of vibration and levels of inclines on gait characteristics and respective variabilities could be observed. Specifically, the supra-threshold vibration might have a stronger effect than sub-threshold/no vibrations during downhill (challenging tasks) walking.

## MATERIALS AND METHODS

### Participants

A total of 18 healthy adults attended this study (eight males and 10 females, average age: 24.05 ± 1.59 years old). The average height of these healthy young adults was 1.72 ± 0.09 m, and the average weight was 67.83 ± 12.45 kg. Also, the average preferred walking speed (PWS) was 1.08 ± 0.17 m/s. Participants were free from any musculoskeletal deficits and had no history of any surgeries in their lower extremities. Participants were excluded if they self-reported any sensory or neurological impairments. This study obeyed the guideline and regulations of the University of Nebraska Medical Center Institutional Review Board that approved this study (IRB# 0338-17-FB). Participants were required to sign an informed consent document before the data collection began. The sample size of this study was selected based on two methods: (1) our previous publication, which was to investigate the effect of different plantar vibrations on obstacle negotiation among healthy young adults (*Song et al., 2022*), and (2) the computation of power analysis through G*Power (http://www.gpower.hhu.de/). For method #1, the effect size was large when recruiting 19 healthy participants (*Song et al., 2022*). For method #2, the $\eta^2 = 0.138$ for the large effect size based on the partial eta squared method was used to calculate the effect size f (*Cohen, 1988*). After obtaining the effect size f = 0.4, the sample size can be estimated. By this calculation, recruiting a total of 18 healthy young participants reached 90% power for using the repeated measure in the current study.

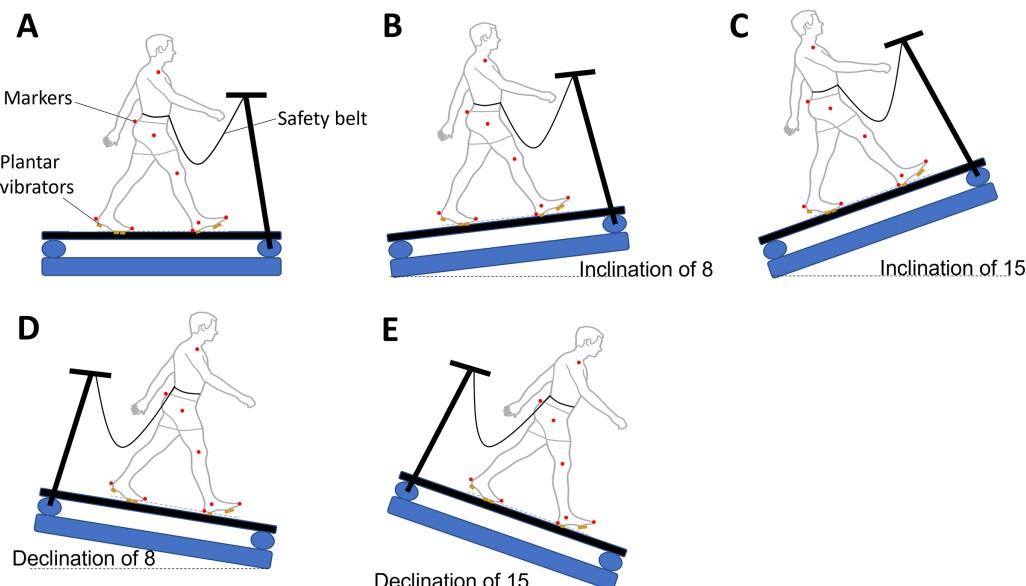

**Figure 1 Five treadmill walking conditions used in this study.** A safety lanyard was attached to participants to ensure the safety. Red dots indicate the retro-reflective markers. (A) Level surface; (B) inclination of eight degrees; (C) inclination of 15 degrees; (D) declination of eight degrees; (E) declination of 15 degrees.

## Experimental protocol

First, the PWS of each participant needed to be identified. Participants experienced walking on five different inclines to determine the PWS, which was kept the same throughout the data collection. The belt ran at 0.8 m/s while participants stepped on this treadmill with holding the handrail. Then, participants were instructed to walk naturally without holding the handrail. At the same time, experimenters evaluated the walking speed as "is this walking speed comfortable as you walk on the street?" The treadmill speed was increased or decreased by 0.1 m/s based on the participants' responses. This step was repeated until the participants confirmed the PWS. Finally, participants needed to select the most comfortable PWS from walking on five different inclines. Once the PWS was confirmed, participants needed to walk 5-min on the level treadmill for familiarization. After the familiarization, a 2-min mandatory rest was assigned to catch up on the shortness of breath. Then, 15 walking trials were randomly assigned to participants (three types of vibration—no/sub/supra-threshold vibrations × five inclines—15%, 8%, 0%, −8%, −15% of grade inclines) (Fig. 1). Each walking trial lasted 2 min. Also, participants walked at the same PWS regardless of what inclines were given. Participants were asked to take a 2-min mandatory rest to wash out the learning effect from any treadmill walking between two trials (*Hu & Chien, 2021*). The rationale that PWS was not controlled because participants recruited in this study had different heights and weights, and they had different natural PWSs. In particular, a previous study had proven that gait differences under walking on fixed-speed treadmill walking were not considered to be clinically meaningful (*Sloot, van der Krogt & Harlaar, 2014*). As we aimed to simulate the natural walking, PWS was selected based on participants' responses. Also, in this study, the effect of inclines and

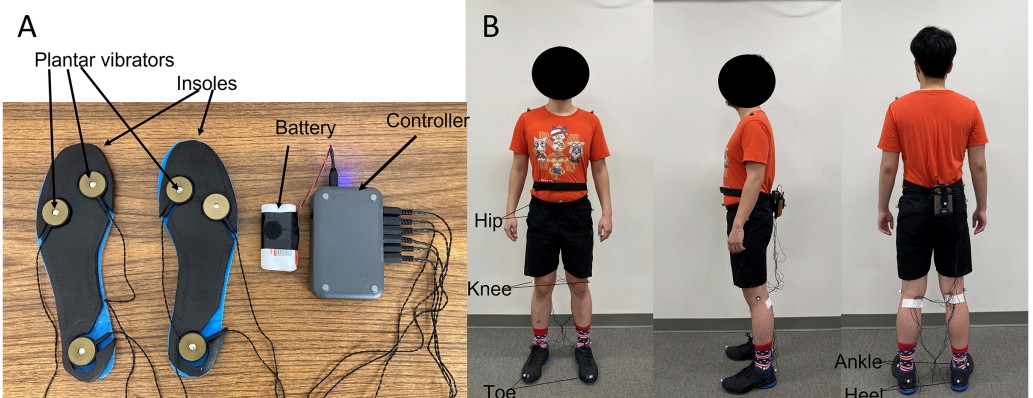

**Figure 2 The plantar vibration device used in this study.** (A) The customized insoles and C2 vibrotactile with a controller and battery; (B) a participant with the plantar vibration device and 12 retro-reflective markers.

the effect of different types of vibrations were compared within subjects, not between groups. Therefore, we believed different walking speeds might not become a confounding factor that affected our results. However, the average walking speed of the present study was slightly slower than the speed reported from a previous review (ranging from 1.00–1.30 m/s) in young adults when walking on a level surface (*Fukuchi, Fukuchi & Duarte, 2019*). Due to the self-selected PWS under five different inclined walking, participants may intentionally select slower PWS for safety when walking downhill. Regardless, this study's average walking speed was still within the reasonable range of average walking speed from another previously reported study (*Murtagh et al., 2021*).

## Experimental materials

A motion capture system with eight cameras was used to record the three-dimensional motion data at 100 Hz (Qualisys AB, Gothenburg, Sweden). A total of 12 retro-reflective markers were attached on the anterior superior iliac spines, greater trochanter of the femur, lateral epicondyle of the femur, lateral malleoli, toe (second metatarsophalangeal ray), and heel of both legs (*Lu, Xie & Chien (2022)*; *Chien, Post & Siu, 2018*). The C2 vibrotactile (Engineering Acoustics, Casselberry, FL, USA) was embedded in customized insoles (Fig. 2). This C2 vibrotactile could reach a frequency of 200–300 Hz and amplitude of 0–22.5 db range resonance by a linear actuator with a moving magnet design. This range of frequency and amplitude has been proven to be able to stimulate the Pacinian corpuscle, the most sensitive of mechanoreceptors (*Song et al., 2022*). Prior to data collection, each participant was required to stand still with bilateral feet on the ground to determine the designated sensory threshold (the minimum amplitude that participants could perceive). The amplitude of vibrotactile was gradually adjusted from 0db using TAction Creator (Engineering Acoustics, Casselberry, FL, USA) until participants could perceive the vibration. Then the designated sensory threshold of each participant was defined (*Severini & Delahunt, 2018*). For supra-threshold plantar vibration, the frequency-amplitude was set 250 Hz and 130% of the designated sensory threshold (*Severini & Delahunt, 2018*). In contrast, for sub-threshold plantar vibration, the frequency-amplitude

was set 250 Hz and 90% of the designated sensory threshold (*Priplata et al., 2003*; *Priplata et al., 2006*). The same brand (Champion Cross Trainer) running shoes with sizes 6–12 were provided for all participants. For each walking trial, after participants constantly walked on the treadmill for 10 s, the plantar vibrotactile was activated at the 11th second. The active duration of the vibrotactile was set at 0.3 s, and the rest duration of the vibrotactile was set at 0.6 s (*Chien et al., 2017*) to prevent saturation of the plantar sensation. All participants needed to wear their own socks, and the insole was for one-time use. After each data collection, all equipment was wiped with a Lysol disinfectant wiper. A treadmill (Biodex RTM 600; Biodex Medical Systems, Inc., Shirley, NY, USA) with a safety lanyard was used for performing human walking. Specifically, this treadmill has an inverse running motor; therefore, downhill walking could be performed. The heel strike was defined when the horizontal heel displacement reached a maximum (*Parks, Chien & Siu, 2019*). The stride time was the period between one contralateral and the next contralateral heel-strike. The stride length was the distance from the contralateral and the next contralateral heel-strike. The step width was the width between heel markers from one ipsilateral to another contralateral heel-strike. In this study, the spatial-temporal parameters for a total of 100 consecutive gait cycles were used (Fig. 1) (*Kroneberg et al., 2019*). These 100 consecutive gait cycles were selected from the 11[th] to the 110[th] stride to eliminate the large step-to-step fluctuations during the beginning/end of walking. The variability was defined using the coefficient of variation.

## Statistical analysis

A Shapiro-Wilk Normality Test was used to test the normality of each dependent variable, with the alpha value set at 0.05. If the alpha value of Shapiro-Wilk was greater than 0.05, a two-way repeated ANOVA (five inclines × three types of vibrations) was used to investigate the effect of different inclines (+15%, +8%, 0%, −8%, −15% grade), the effect of vibrations, and the interaction between these two parameters on each dependent variable. The dependent variables, based on previous literature, included stride length, stride time, step width, and their variabilities (*Chien, Post & Siu, 2018*). If an interaction existed in a dependent variable, *post hoc* comparisons were performed using the Tukey method. The significance level was set at 0.05. Statistical analysis was completed in SPSS 20.0 (IBM Corporation, Armond, NY, USA). If the alpha value of Shapiro-Wilk was smaller than 0.05, a Friedman test was used. Wilcoxon Signed-Rank test was used for pairwise comparisons for each dependent value. To understand the effect size, we used the Partial Eta Squared method. This method is widely used to measure the effect size, noting based on *Cohen (1988)* guideline that 0.138 for a large effect size, 0.059 for a moderate effect size, and 0.01 for a small effect size.

## RESULTS

The alpha values of the Shapiro-Wilk Test for each dependent variable were greater than 0.05, indicating that the data were normally distributed. Detailed means and standard deviations for dependent variables are shown in Table 1. A significant interaction was observed in the stride time ($F_{8,136} = 11.34$, $p < 0.001$) and the step width ($F_{8,136} = 2.48$,

**Table 1 Summary of gait characteristics data with different types of plantar vibration on different inclines for all participants.**

| | Gait characteristics | | | | |
|---|---|---|---|---|---|
| Conditions | Step time (s) | Stride time (s) | Step length (m) | Stride length (m) | Step width (m) |
| **Up15** | | | | | |
| No | 0.598 (0.071) | 1.198 (0.141) | 0.320 (0.052) | 1.540 (0.215) | 0.177 (0.031) |
| Sub | 0.600 (0.067) | 1.200 (0.134) | 0.322 (0.054) | 1.547 (0.215) | 0.179 (0.031) |
| Supra | 0.597 (0.064) | 1.193 (0.128) | 0.314 (0.049) | 1.540 (0.212) | 0.178 (0.032) |
| **Up8** | | | | | |
| No | 0.595 (0.056) | 1.191 (0.113) | 0.309 (0.048) | 1.540 (0.201) | 0.186 (0.035) |
| Sub | 0.598 (0.058) | 1.196 (0.118) | 0.304 (0.053) | 1.545 (0.206) | 0.194 (0.038) |
| Supra | 0.593 (0.064) | 1.188 (0.126) | 0.302 (0.053) | 1.532 (0.210) | 0.184 (0.036) |
| **Level** | | | | | |
| No | 0.578 (0.049) | 1.158 (0.099) | 0.270 (0.051) | 1.487 (0.182) | 0.201 (0.047) |
| Sub | 0.575 (0.044) | 1.149 (0.088) | 0.257 (0.051) | 1.482 (0.195) | 0.213 (0.036) |
| Supra | 0.574 (0.046) | 1.149 (0.092) | 0.263 (0.056) | 1.486 (0.196) | 0.209 (0.038) |
| **Down8** | | | | | |
| No | 0.556 (0.039) | 1.111 (0.078) | 0.565 (0.084) | 1.391 (0.185) | 0.126 (0.031) |
| Sub | 0.555 (0.042) | 1.110 (0.085) | 0.565 (0.082) | 1.392 (0.181) | 0.121 (0.027) |
| Supra | 0.555 (0.041) | 1.110 (0.082) | 0.564 (0.085) | 1.390 (0.186) | 0.120 (0.025) |
| **Down15** | | | | | |
| No | 0.533 (0.038) | 1.067 (0.076) | 0.532 (0.076) | 1.308 (0.166) | 0.130 (0.030) |
| Sub | 0.534 (0.038) | 1.067 (0.077) | 0.564 (0.085) | 1.310 (0.166) | 0.125 (0.029) |
| Supra | 0.532 (0.037) | 1.064 (0.074) | 0.531 (0.075) | 1.307 (0.165) | 0.127 (0.027) |

**Note:**

Data are shown as the Mean (SD). Down8, Declination of 8 degrees; Down15, Declination of 15 degrees; Level, level surface; Up8, Inclination of 8 degrees; Up15, Inclination of 15 degrees; Sub, with sub-threshold plantar vibration; Supra, with supra-threshold plantar vibration; No, without plantar vibration.

$p = 0.015$). *Post hoc* pair-comparisons showed that applying the supra-threshold plantar vibration during downhill walking decreased the stride time compared to applying no plantar vibration ($-8\%$: $p < 0.001$, $-15\%$: $p < 0.001$) and applying the sub-threshold plantar vibration ($-8\%$: $p < 0.001$, $-15\%$: $p < 0.001$). Also, the results showed that applying the sub-threshold plantar vibration increased the step width during uphill walking compared to applying no ($8\%$: $p = 0.036$) and supra-threshold plantar vibration ($8\%$: $p < 0.001$). More details are shown in Figs. 3 and 4.

A significant effect of vibration was found in stride time ($F_{2,34} = 63.65$, $p < 0.001$), stride time variability ($F_{2,34} = 3.368$, $p = 0.046$), and stride length variability ($F_{2,34} = 3.733$, $p = 0.034$). The results of marginal means indicated that applying supra-threshold plantar vibration significantly decreased the stride time more than applying no ($p < 0.001$) or sub-threshold plantar vibration ($p < 0.001$). It also indicated that applying the supra-threshold plantar vibration decreased the stride time variability ($p = 0.028$) and the stride length variability ($p = 0.017$) compared to applying the sub-threshold plantar vibration. More details are shown in Table 2 and Fig. 5.

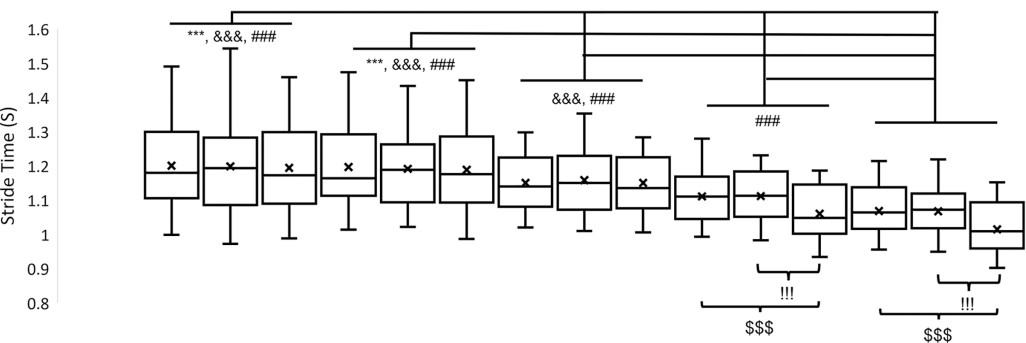

**Figure 3 Stride time in 15 different walking conditions.** From left to right, the walking conditions are uphill 15%_sub-threshold, uphill 15%_no, uphill 15%_supra-threshold, uphill 8%_sub-threshold, uphill 8%_no, uphill 8%_supra-threshold, level_sub-threshold, level_no, level_supra-threshold, downhill 8%_sub-threshold, downhill 8%_no, downhill 8%_supra-threshold, downhill 8%_sub-threshold, downhill 8%_no, downhill 8%_supra-threshold, respectively. Error bars indicate between-participant standard deviations. X indicates the mean of stride time. The symbols *, #, and & indicate the significant difference among inclines; ! and $ indicate the significant difference among vibration conditions.

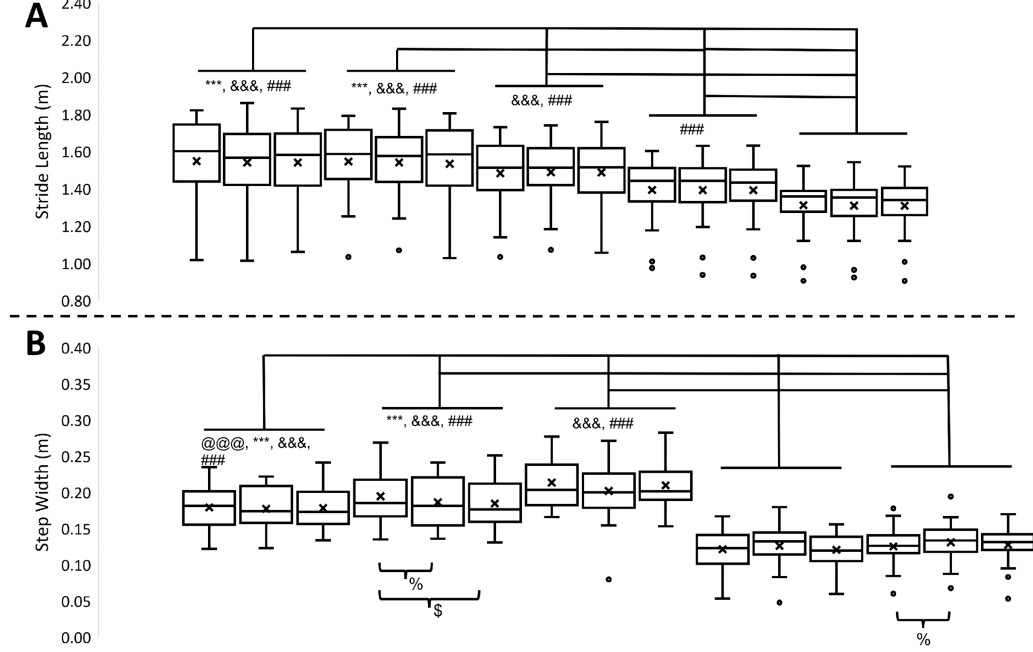

**Figure 4 Spatial variables in 15 different walking conditions.** (A) Stride length in 15 different walking conditions. (B) Step width in 15 different walking conditions. From left to right, the walking conditions are uphill 15%_sub-threshold, uphill 15%_no, uphill 15%_supra-threshold, uphill 8%_sub-threshold, uphill 8%_no, uphill 8%_supra-threshold, level_sub-threshold, level_no, level_supra-threshold, downhill 8%_sub-threshold, downhill 8%_no, downhill 8%_supra-threshold, downhill 8%_sub-threshold, downhill 8%_no, downhill 8%_supra-threshold, respectively. Error bars indicate between-participant standard deviations. X indicates the mean of stride time. Black dots indicate the outlier. The symbols *, #, @, and & indicate the significant difference among inclines, and % and $ indicate the significant difference among vibration conditions.

**Table 2 The effect of vibration, inclination, and the interaction of vibration and inclination on gait variabilities.**

| | Mean (SD) | Vibrations | Inclines | vibrations × inclines |
|---|---|---|---|---|
| **(A) Stride time variability (%, second/second)** | | | | |
| Up15_Sub | 2.49 (0.90) | $p = 0.046$ | $p < 0.001$ | NS |
| Up15_No | 2.49 (1.08) | | | |
| Up15_Supra | 2.29 (0.88) | **Marginal Means** | **Marginal Means** | |
| Up8_Sub | 2.48 (1.09) | Supra *vs* Sub: $p = 0.028$ | Level *vs* Up15: $p < 0.001$ | |
| Up8_No | 2.18 (0.83) | | Level *vs* Up8: $p < 0.001$ | |
| Up8_Supra | 2.19 (0.86) | | Level *vs* Down8: $p = 0.015$ | |
| Level_Sub | 1.78 (0.89) | | Level *vs* Down15: $p = 0.015$ | |
| Level_No | 1.79 (0.78) | | Up15 *vs* down8: $p = 0.004$ | |
| Level_Supra | 1.67 (0.69) | | Up15 *vs* down15: $p = 0.028$ | |
| Down8_Sub | 2.03 (0.80) | | | |
| Down8_No | 2.04 (0.69) | | | |
| Down8_Supra | 1.98 (0.87) | | | |
| Down15_Sub | 2.07 (0.52) | | | |
| Down15_No | 2.07 (0.59) | | | |
| Down15_Supra | 2.14 (0.82) | | | |
| **(B) Stride length variability (%, meter/meter)** | | | | |
| Up15_Sub | 2.61 (0.92) | $p = 0.046$ | $p < 0.001$ | NS |
| Up15_No | 2.70 (1.14) | | | |
| Up15_Supra | 2.42 (0.94) | **Marginal Means** | **Marginal Means** | |
| Up8_Sub | 2.60 (1.19) | Supra *vs* Sub: $p = 0.017$ | Level *vs* Up15: $p < 0.001$ | |
| Up8_No | 2.28 (0.84) | | Level *vs* Up8: $p < 0.001$ | |
| Up8_Supra | 2.32 (0.91) | | Level *vs* Down8: $p < 0.001$ | |
| Level_Sub | 1.92 (0.92) | | Level *vs* Down15: $p < 0.001$ | |
| Level_No | 1.99 (0.87) | | Up15 *vs* Up8: $p = 0.029$ | |
| Level_Supra | 1.86 (0.73) | | Up15 *vs* Down8: $p < 0.001$ | |
| Down8_Sub | 2.17 (0.97) | | Up8 *vs* Down8: $p = 0.031$ | |
| Down8_No | 2.23 (0.72) | | | |
| Down8_Supra | 2.14 (0.89) | | | |
| Down15_Sub | 2.38 (0.60) | | | |
| Down15_No | 2.31 (0.67) | | | |
| Down15_Supra | 2.39 (0.72) | | | |
| **(C) Step width variability (%, meter/meter)** | | | | |
| Up15_Sub | 10.61 (3.21) | NS | $p < 0.001$ | NS |
| Up15_No | 10.49 (3.58) | | | |
| Up15_Supra | 10.28 (3.46) | | **Marginal Means** | |
| Up8_Sub | 10.79 (3.91) | | Level *vs* Up15: $p = 0.002$ | |
| Up8_No | 10.48 (4.15) | | Level *vs* Up8: $p < 0.001$ | |
| Up_Supra | 10.34 (3.38) | | Level *vs* Down8: $p < 0.001$ | |
| Level_Sub | 8.8 (3.21) | | Level *vs* Down15: $p < 0.001$ | |

|  | Mean (SD) | Vibrations | Inclines | vibrations × inclines |
|---|---|---|---|---|
| Level_No | 9.46 (4.48) |  | Up15 *vs* Down15: $p < 0.001$ |  |
| Level_Supra | 8.98 (4.41) |  | Up15 *vs* Down8: $p < 0.001$ |  |
| Down8_Sub | 21.05 (11.17) |  | Up8 *vs* Down8: $p < 0.001$ |  |
| Down8_No | 22.01 (12.38) |  | Up8 *vs* Down15: $p < 0.001$ |  |
| Down8_Supra | 21.36 (10.26) |  |  |  |
| Down15_Sub | 22.85 (10.52) |  |  |  |
| Down15_No | 21.88 (9.25) |  |  |  |
| Down15_Supra | 21.9 (9.71) |  |  |  |

**Note:**

(A) Stride time variability; (B) Stride length variability; (C) Step width variability. Down8, Declination of 8 degrees; Down15, Declination of 15 degrees; Level, level surface; Up8, Inclination of 8 degrees; Up15, Inclination of 15 degrees; Sub, with sub-threshold plantar vibration; Supra, with supra-threshold plantar vibration; No, without plantar vibration; NS, not significant.

A significant effect of inclines was observed in stride time ($F_{4,68} = 58.334$, $p < 0.001$), stride time variability ($F_{4,68} = 15.570$, $p < 0.001$), stride length ($F_{4,68} = 108.172$, $p < 0.001$), stride length variability ($F_{4,68} = 15.548$, $p < 0.001$), step width ($F_{4,68} = 60.321$, $p < 0.001$), and step width variability ($F_{4,68} = 34.814$, $p < 0.001$). For all variabilities, the results of marginal means indicated that a U shape in the stride time variability, the stride length variability, and the step width variability, indicating that no matter increasing or decreasing the inclines from level, the variabilities increased. More details are shown in Figs. 3–5, and Table 2.

The effect size was large because the Partial Eta Squared values for the interaction between the effect of vibrations and the effect of inclines were 0.4 for the stride time and 0.127 for the step width based on the literature (*Cohen, 1988*).

# DISCUSSION

This study attempted to investigate the interaction between the effect of different inclines and the effect of different types of plantar vibration in gait characteristics and respective variabilities among healthy young adults. The results partially agreed with our hypothesis that a significant interaction was found in the stride time and step width. Specifically, the supra-threshold plantar vibration decreased the stride time during downhill walking, and the sub-threshold plantar vibration increased the step width during uphill walking. However, the interaction didn't reach the levels of significance in the gait variabilities.

## The different types of vibration affected gait characteristics differently depending on the different locomotor tasks

Unsurprisingly, the decrement in stride time was observed when walking downhill than walking level or uphill in the current study. This decrement in stride time without changing stride length led to increased cadence. In other words, the frequency of the foot staying on the ground increased. The main purpose of this typical gait pattern was to

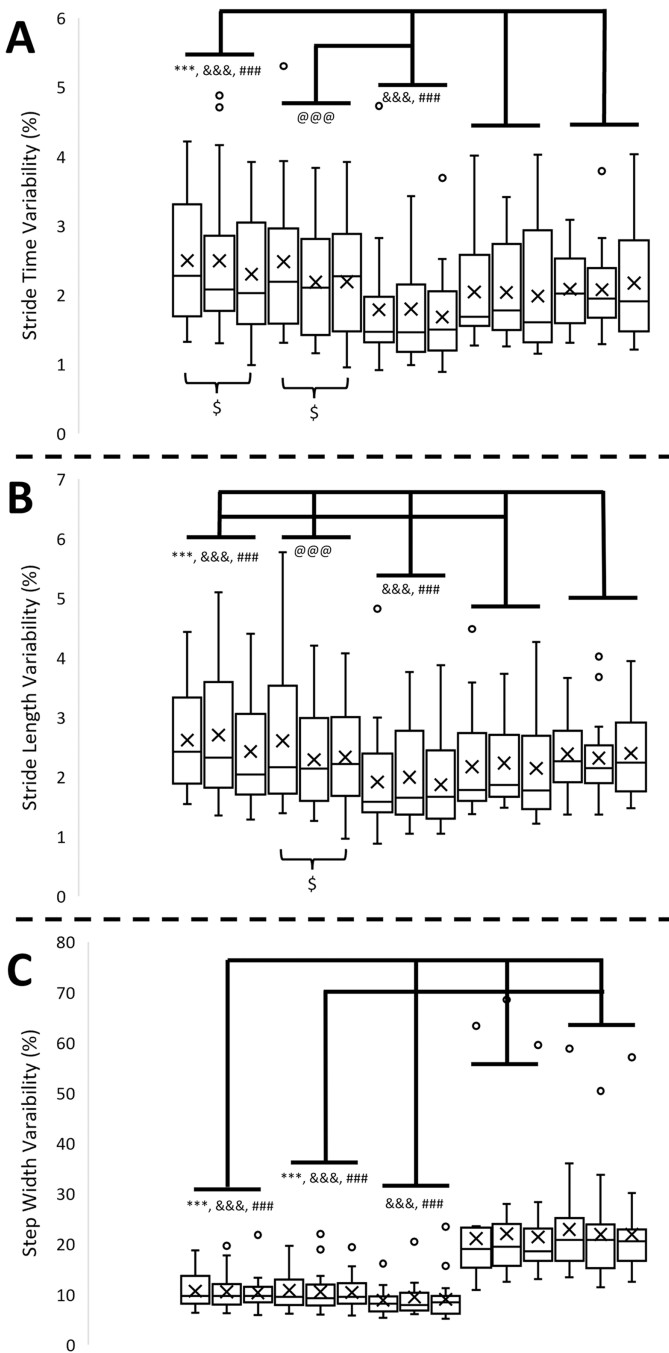

**Figure 5 Gait variabilities in 15 different walking conditions.** (A) Stride time variability in 15 different walking conditions. (B) Stride length variability in 15 different walking conditions. (C) Step width variability in 15 different walking conditions. From left to right, the walking conditions are uphill 15%_sub-threshold, uphill 15%_no, uphill 15%_supra-threshold, uphill 8%_sub-threshold, uphill 8%_no, uphill 8%_supra-threshold, level_sub-threshold, level_no, level_supra-threshold, downhill 8%_sub-threshold, downhill 8%_no, downhill 8%_supra-threshold, downhill 8%_sub-threshold, downhill 8%_no, downhill 8%_supra-threshold, respectively. Error bars indicate between-participant standard deviations. X indicates the mean of stride time. Black dots indicate the outlier. The symbols *, #, and & indicate the significant difference among inclines, and $ indicates the significant difference among vibration conditions.

ensure dynamic balance during walking (*Williams, Peterson & Earhart, 2013*). We speculated the fear of falling might be the major factor causing this typical gait pattern (*Kimel-Naor, Gottlieb & Plotnik, 2017*). Although this study did not quantify the fear of falling, these young adults were described verbally as "awkward and feared stepping over the treadmill feelings" after walking downhill. Interestingly, a further decrease in stride time was observed during downhill walking in the current study while applying the supra-threshold plantar vibration than other types of vibration. The rationale might be that the supra-threshold plantar vibration played a role in perturbation (*Peters et al., 2020*; *Song et al., 2022*, *Pathak et al., 2022*). In *Peters et al. (2020)* study, applying the supra-threshold vibration on the medial forefoot induced the latency of muscle responses. Also, applying the supra-threshold on the plantar area increased the toe clearance during obstacle negotiation (*Song et al., 2022*) and increased the minimum toe clearance variability during level walking (*Pathak et al., 2022*). These abovementioned alternations in muscle activations and gait patterns were suggested as signs when encountering the disruptive dynamic balance situation due to the temporary loss of sensation from the plantar area (*Chien et al., 2017*). Specifically, a study used the electroencephalographic method to assess brain activations under short-term supra-threshold vibration on the Achilles tendon (*Zinke et al., 2019*). The results indicated that short-term local Achilles tendon supra-vibration induced lower brain activations, specifically the alpha-1 and beta-1 band, inferring the process of re-mapping between the somatosensory cortex activations and the anticipatory muscle responses (*Buchholz, Jensen & Medendorp, 2014*). Therefore, it could be reasonable to speculate that the young adults in the current study might have no choice but to decrease stride time during downhill walking as an outcome when the gains between the somatosensory cortex and gait characteristics needed to be re-mapped due to the perturbed plantar sensation. Surprisingly, the effect of sub-threshold plantar vibration didn't reach statistical significance while walking downhill. Based on previous studies, applying sub-threshold plantar vibration played a role in enhancing balance control in standing and walking in older fallers (*Aboutorabi et al., 2018*; *Baudry & Duchateau, 2020*), patients with Parkinson's disease (*High et al., 2018*), patients with stroke (*Önal, Karaca & Sertel, 2020*) and patients with the diabetic disorder (*Nelander et al., 2012*). However, applying sub-threshold vibration doesn't enhance balance control in walking and standing on a level surface in young adults (*Acuña, Zunker & Thelen, 2019*). Similarly, this current result concluded that sub-threshold plantar vibration might not affect gait characteristics during downhill walking.

When healthy young adults performed uphill walking, the different effect of plantar vibration on gait characteristics was observed. It has been shown that walking downhill requires more energy than walking on a level surface or uphill (*Minetti et al., 2002*). Also, a preliminary result related to brain activities indicated that the oxygenation level was more significant in the prefrontal and sensorimotor cortex areas when walking on a downhill surface *vs* walking on a level or an uphill surface (*Mazerie et al., 2012*). Thus, in comparison to walking downhill, applying supra-threshold plantar vibration in these young adults while walking uphill might not perturb their gait characteristics because uphill walking might not be as challenging as downhill walking. Another interesting

finding was that while walking either downhill or uphill, the step width decreased compared to level walking. The significant increases or decreases in step width from the healthy condition were associated with the safety-concerned gait (*Brach et al., 2005*). In the current study, these significant decreases from marginal means in step-width during up/downhill walking might also be the safety-concerned gait in these young participants. While taking a close look at the effect of plantar vibration, applying sub-threshold plantar vibration increased the step width when walking on the grade of 8% uphill compared to other conditions. The rationale might be that applying the sub-threshold plantar vibration at this specific incline eases the safe-concerned gait by increasing the base of support (*Likens et al., 2020*; *Song et al., 2022*).

Surprisingly, the effect of sub-threshold vibration was only observed while walking 8% grade uphill. It might be the different inclines induced different control mechanisms. These speculations were supported by Earhart and Bastian that healthy young participants used different control mechanisms when they stepped on steeper grades of a wedge (*Earhart & Bastian, 2000*). Similarly, another study, which investigated the first step on an inclined surface, supported the above observation that the control mechanisms were changed to prepare the limb for an elevated heel contact with increased propulsive force when the grade of inclination reached approximately 11% (*Prentice et al., 2004*). Thus, due to the change in the control mechanism, the effect of sub-threshold vibration might be eliminated. To sum up, this study was the first to identify that sub- and supra-threshold vibration affected the gait characteristics differ depending on the different locomotor tasks.

### The gait variability measure failed to identify the interaction between the effect of different inclines and the effect of different types of vibrations

In the past decades, gait variability has been treated as a critical indicator for identifying the gait characteristics in various diseases related to aging or neurological impairment (*Ma et al., 2020*). Also, an increase in gait variability has been thought to be a sign of increased stride-to-stride fluctuations. In the current study, the interaction between the effect of different inclines and the effect of different types of vibrations was not found in the step time variability, the step length variability, and the step width variability. From the statistical point of view, the current results meant that the effect of vibration was constant across all levels of the impact of inclines. Thus, the further main effect of different types of vibrations were investigated. The marginal means indicated that applying the supra-threshold plantar vibration significantly reduced both stride time variability and stride length variability compared to applying the sub-threshold plantar vibrations on locomotion tasks. This study proposed an explanation for this phenomenon that applying supra-threshold vibration stimulated the plantar area rhythmically. Specifically, for supra-threshold plantar vibration, which participants could perceive, this cue became guidance for these young adults to follow, resulting in less fluctuation of stride time and stride length. However, why did the supra-threshold plantar vibration perturb the plantar sensation but still decrease the variability, which indicates fewer gait fluctuations? It could be explained by optimal variability theory, in which too much or too little gait variability

compared to normal walking leads to instability of gait (*Brach et al., 2005*). For the effect of different inclines, a U shape was found in stride time variability, stride length variability, and step width variability, like the previous study (Fig. 5) (*Castano & Huang, 2021*). These results had an agreement that more difficult locomotor tasks induced larger stride-to-stride fluctuations. Nevertheless, the gait variability measure failed to identify the interaction between the effect of different inclines and the effect of different vibrations.

## Clinical implications

A previous study suggested that vibration enhanced the gait velocity in stroke survivors after a 6-week intervention (*Lee, 2019*). Additionally, a review involving 2,658 patients with stroke indicated that stroke survivors who received level treadmill intervention were not likely to improve their ability to walk independently than those who did not receive the treadmill training (*Mehrholz et al., 2013*). This review further suggested that increasing training intensities might be required, such as walking on the inclines (*Mehrholz et al., 2013*). However, to our best knowledge, there was no previous study combining plantar vibration with inclined walking as a rehabilitation protocol to improve gait performance. Although only healthy young adults were recruited in this study, the potential clinical benefits of applying plantar vibration combined with inclined walking to enhance dynamic balance function were observed. Specifically, based on the observations from this study, the sub-threshold plantar vibration combined with uphill treadmill training may be suggested for the early stage of rehabilitation in patients with acute stroke. With progression in rehabilitation, high-intensity treadmill training (downhill with supra-threshold vibrations) may further be suggested to enhance the independence of walking in patients with stroke with an assistant device.

## Limitations and future studies

The apparent limitation of the current study was the relatively small sample size. Nevertheless, based on the measurement of Partial Eta Squared, the effect size in the current study was moderate to large, indicating the practical significance of this research finding. Another limitation was that we did not record the designated sensory threshold of each participant. It might be inevitable that different individuals demonstrate different responses to the same percentage of vibration amplitude. However, in this current study, our major aim was to investigate the interaction between the effect of inclines and the effect of different types of vibrations. Therefore, the designated sensory thresholds were selected (90% of designated sensory thresholds as the sub-threshold vibration and 130% of designated sensory thresholds as the supra-threshold vibration) based on limited previous studies. Importantly, these studies agreed that these two types of vibrations were sufficient to impact the gait characteristics (*Severini & Delahunt, 2018*). A future study is needed to resolve this limitation by performing a correlation analysis between the designated sensory threshold and the gait characteristics of each individual. Besides, only healthy young adults were recruited in this study. Will older adults, older fallers, patients with musculoskeletal disorders, or patients with neurological disorders have similar responses to different types of vibration during different inclines walking? Our team will attempt to investigate all

these important research questions in the near future. In order not to obscure the focus on the outcomes from conventional linear statistics in gait characteristics, the levels of complexity, the bifurcations toward chaos, and the regularity of gait didn't be analyzed in the current study. Future studies will re-analyze the abovementioned outcomes under the effect of different types of plantar vibration and the effect of different inclines. Based on previous studies, we expect to observe that increasing the inclines would extend the critical values in the first period-doubling routes to chaos and increase the regularity of gait. Notably, the reduction in the complexity of gait after implementing the plantar vibrations might be observed, resulting in the enhancement of the gait.

## CONCLUSIONS

In summary, the sub- and supra-threshold plantar vibration demonstrated different effects on the gait characteristics and variabilities in different inclines. The supra-threshold vibration played the role of somatosensory perturbation and made young adults decrease their stride time, particularly during downhill walking, than walking on other inclines. On the other hand, the sub-threshold vibrations increased the step width during uphill walking. The current findings may develop a foundation for future use in combination with different types of vibration and different levels of inclined treadmill intervention in stroke survivors or patients with other neurological disorders.

## ACKNOWLEDGEMENTS

We would like to thank all participants for their contribution to the study. All data collection were performed at the Clinical Movement Analysis Lab, Department of Health & Rehabilitation Science at the University of Nebraska Medical Center. We sincerely thank the generosity of the Department of Health & Rehabilitation Science for using the equipment.

### Funding

The authors received no funding for this work.

### Competing Interests

The authors declare that they have no competing interests.

### Author Contributions

- Haoyu Xie performed the experiments, analyzed the data, prepared figures and/or tables, authored or reviewed drafts of the article, and approved the final draft.
- Haolan Liang performed the experiments, prepared figures and/or tables, and approved the final draft.
- Jung H. Chien conceived and designed the experiments, performed the experiments, analyzed the data, prepared figures and/or tables, authored or reviewed drafts of the article, and approved the final draft.

## Human Ethics

The following information was supplied relating to ethical approvals (*i.e.*, approving body and any reference numbers):

The Institutional Review Board of the University of Nebraska Medical Center approved this study (IRB# 338-17-FB).

## Data Availability

The raw measurements are available in the Supplemental File.

## Supplemental Information

Supplemental information for this article can be found online at http://dx.doi.org/10.7717/peerj.14619#supplemental-information.

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
