# Peer review of "Different types of plantar vibration affect gait characteristics differently while walking on different inclines"

_PeerJ, doi:10.7717/peerj.14619_

## Round 0.1 · original submission · Major Revisions

Some major changes are needed. Please complete with the necessary information.

Best regards,

·

Basic reporting

According to the analyzed research, the available evidence and findings will fill the existing gap in the knowledge in the field.

Experimental design

Research is within the aims and scope of the journal. Methods are described with sufficient detail and information for replication.

Validity of the findings

The aim was to investigate the interaction between the effect of inclines and the effect of different types of vibrations. The apparent limitation of the current study was the relatively small sample size.

Additional comments

Ln42 – “Nicola Tesla” – Nikola Tesla.
Ln190 – “...significant level...” – Please consider rewriting as “significance level.”
Ln361-367 – “A previous study had suggested……..on the inclines (Mehrholz et al., 2013).” - Please keep in mind that previous studies should be discussed within the discussion section, not in the conclusion. In conclusion, only the study's main findings (and practical application) should be presented.
Figure 1 - There is no (A), (B), (C),… visible within the figure. Please recheck all figures and add the symbols where needed.
Table 1
2.49 (0.90) – It is clear that this presents the mean value and standard deviation. However, there is no Unit of measure visible in the table. Please add the unit of measure everywhere, wherever necessary.

·

Basic reporting

Thank you for submitting the manuscript entitled, “Different types of plantar vibration affect gait characteristics while walking on different inclines’’. The area of the research is very interesting; however, it needs a few amendments. Overall, the paper is well-written.

The manuscript is well-structured, and it is easy to follow the sections.

The key words should be different from the words in the title.

The introduction provides a good background of the topic, except for the part that it mentions a lot about patients with different affections and only a few about healthy young adults, who, in fact, are the subjects of this study.

The article innovation should be more clearly presented in the Introduction.

Figures are relevant and help the reader to understand the manuscript.

The quality of the images is good enough, but I don’t know if the reviewing version has lower resolution than the final version. If not, images should have better resolution in its final size.

Experimental design

The experimental design meets the scope of the journal, and the hypothesis is well-defined and it’s relevant to the science community.
Methods are described detailed enough.
The sample size of participants looks rather small. Was there a priori power analysis to establish the sample size?

Validity of the findings

Most of the results are quite interesting and are well discussed.
The reader should be informed about the results following the 3 types of vibration used on each incline.

In the Discussion it would be better to have seen more use of terms like 'originality' and 'significance'. Identify what is new in this study that may benefit readers or how it may advance existing knowledge or create new knowledge on this subject. There should be a clear conclusion on why the research findings are significant.

Reviewer 3 ·

Basic reporting

This paper deals with the investigation from statistical point of view of the human locomotion upon and down inclined planes. Authors presented different results showing the importance of the walking on inclined and declined planes and its benefits.

Experimental design

The results and outputs are relevant by they can be improved.

Also how you vary the speed of walking and how can be controlled? Is the human locomotion can remain at the same speed while walking upon or down an inclined plane?

I suggest adding more graphical results instead of tables. They are more readable than tables.

Validity of the findings

Actually, there are bipedal robots with simple structures, like the compass-gait biped robot, the torso-driven biped robot, the biped robot with knees, that can be used in order to study the human locomotion while walking upon and down slopes. Further discussion in the introduction can be added to enrich it.

Additional comments

It has been shown in the literature that the bipedal/human walking can show complex behaviors like the chaotic gaits, and not only simple or periodic behaviors. These complex behaviors can be exhibited by varying some parameters like the slope angle, the desired speed, the length of legs, and so on.

Further comments about this on how complex motions can appear in the presented results and generally in the locomotion upon and down inclined sloped

---

## Round 0.2 · accepted · Accept

Your paper has been accepted for publication.

·

Basic reporting

Thank you for providing this comprehensive work.
The authors have presented an improved version of the manuscript.

The introduction provides a proper background of the topic.
All the sections have been improved.
Relevant results are well-organised to follow the hypothesis.
The discussion section has been restructured.
The quality of the images is good enough.
It seems that the English is technically correct.

Experimental design

The experimental design meets the scope of the journal, and it is relevant to the community.
Methods are described detailed enough.

Validity of the findings

The results and the conclusions are quite interesting and well-discussed. All data are provided.

The authors have adequately addressed all my comments. I have no further suggestions.

Reviewer 3 ·

Basic reporting

Article well written and structured and commented

Experimental design

The experimental results are clear and sufficient

Validity of the findings

The results and the output seem to be valid